# Modulation of Fatty Acid Composition of *Aspergillus oryzae* in Response to Ethanol Stress

**DOI:** 10.3390/microorganisms7060158

**Published:** 2019-06-01

**Authors:** Long Ma, Lijun Fu, Zhihong Hu, Yongkai Li, Xing Zheng, Zhe Zhang, Chunmiao Jiang, Bin Zeng

**Affiliations:** Jiangxi Key Laboratory of Bioprocess Engineering, College of Life Sciences, Jiangxi Science & Technology Normal University, Nanchang 330013, China; president114@163.com (L.M.); 18702597032@163.com (L.F.); huzhihong426@163.com (Z.H.); lyk2018@jxstnu.edu.cn (Y.L.); zhenxing0504@live.com (X.Z.); zz2007138221@163.com (Z.Z.); jiangcm0810@jxstnu.edu.cn (C.J.)

**Keywords:** *Aspergillus oryzae*, ethanol stress, linoleic acid, fatty acid unsaturation

## Abstract

The koji mold *Aspergillus oryzae* is widely adopted for producing rice wine, wherein koji mold saccharifies rice starch and sake yeast ferments glucose to ethanol. During rice wine brewing, the accumulating ethanol becomes a major source of stress for *A. oryzae*, and there is a decline in hydrolysis efficiency. However, the protective mechanisms of *A. oryzae* against ethanol stress are poorly understood. In the present study, we demonstrate that ethanol adversity caused a significant inhibition of mycelium growth and conidia formation in *A. oryzae*, and this suppressive effect increased with ethanol concentration. Transmission electron microscopy analysis revealed that ethanol uptake triggered internal cellular perturbations, such as irregular nuclei and the aggregation of scattered vacuoles in *A. oryzae* cells. Metabolic analysis uncovered an increase in fatty acid unsaturation under high ethanol conditions, in which a large proportion of stearic acid was converted into linoleic acid, and the expression of related fatty acid desaturases was activated. Our results therefore improve the understanding of ethanol adaptation mechanisms in *A. oryzae* and offer target genes for ethanol tolerance enhancement via genetic engineering.

## 1. Introduction

*Aspergillus oryzae*, commonly known as koji mold, is widely adopted for producing rice wine (sake, makgeolli, and huang-jiu) in East Asia [1] due to its outstanding capability of producing multienzymes such as α-amylase and glucoamylase [2]. The traditional alcoholic beverage of rice wine is produced through parallel fermentation, with a simultaneous saccharification of rice by the koji mold *A. oryzae* and ethanol fermentation by the sake yeast *Saccharomyces cerevisiae* [3]. During the process of saccharification, the starch in steamed rice is continuously hydrolyzed by α-amylase and glucoamylase provided by *A. oryzae*, and this process continues throughout the period of glucose production until the death of *A. oryzae* cells and the autolysis of mycelia [2]. However, during rice wine brewing, the accumulating ethanol in the culture becomes a major source of stress for *A. oryzae*, which ultimately restrains the fermentation performance and reduces hydrolysis efficiency. Therefore, enhancing the ethanol tolerance of *A. oryzae* is important for prolonging fermentation time in rice wine production. Ethanol is well known as an inhibitor of the growth of microorganisms. It has been reported that the accumulation of ethanol affects the integrity of the cell membrane, inactivates cellular enzymes, and inhibits cell growth and viability, leading to cell death during fermentation [4,5]. High levels of ethanol are reported to destroy protein structure, leading to the denaturation of cellular proteins, such as the key glycolytic enzymes pyruvate kinase and hexokinase [6]. To date, the ethanol tolerance mechanisms have been well documented in the budding yeast *S. cerevisiae* [7]. However, the protective mechanism of *A. oryzae* against ethanol stress, which is critical to its survival and adaption to high-ethanol adversity in rice wine brewing, remains ambiguous. 

Many studies have documented that the alteration of cellular lipid composition represents an important factor in adaptation to ethanol perturbation [8,9]. The cellular membrane, the prime barrier between a cell and the environment, is the first target of the injurious effect of ethanol. To counteract an increase in membrane fluidity, many microorganisms respond to ethanol by altering their membrane compositions. In *S. cerevisiae*, it has been reported that yeast cells increase the amount of monounsaturated fatty acids in cellular lipids and incorporate oleic acid into the lipid membrane to counteract the fluidizing effects of ethanol [8,9,10,11]. Similarly, *Bacillus subtilis* and *Clostridium acetobutylicum* elevate the content of saturated fatty acids under ethanol conditions [12]. In addition to unsaturated fatty acids (UFAs), the membrane’s ergosterol contents also contribute to ethanol tolerance, possibly due to their effect on the enhancement of membrane rigidity [13]. Consistent with this assumption, the mutants defective in ergosterol biosynthesis are extremely susceptible to ethanol adversity [14,15].

To cope with the effect of ethanol, heat shock proteins (HSPs) are required to assist with the folding and maintenance of newly assembled proteins and the suppression of protein aggregates [16]. In agreement, the yeast mutants of *HSP12*, *HSP30*, or *HSP104* genes become hypersensitive to ethanol [15,17,18]. To protect yeast cells from protein denaturation caused by ethanol, trehalose cooperates with HSPs to stabilize protein structures through its function in protein binding and the reduction of water activity [19]. Higher intracellular trehalose accumulation through overexpression of trehalose-6-phosphate synthase has enhanced ethanol tolerance in an engineered yeast strain [20]. Moreover, some amino acids have been proven to enhance the ethanol tolerance of a microorganism. Proline has been found to prevent aggregation during protein refolding and to strengthen the membrane and proteins. In yeast, the *pro1* mutant strain that is defective in proline synthesis is more susceptible to ethanol, and conversely, engineered yeast strains with proline accumulation have successfully shown an enhanced tolerance to many stresses, including freezing, desiccation, oxidation, and ethanol [18,21]. A further study has revealed that proline accumulation dramatically reduces the levels of reactive oxygen species (ROS) and increases the survival rate of yeast cells under ethanol conditions [22]. A comparative genomics analysis of five yeast strains with different ethanol tolerances revealed that chromosome III was important to the ethanol stress response, and this aneuploidy contributed to a rapid response to ethanol stress; and conversely, the removal of an extra chromosome III copy in an ethanol-tolerant strain strongly weakened its tolerance [23].

Obviously, an enhancement in the tolerance of *A. oryzae* to ethanol has a great advantage in prolonging fermentation time and increasing hydrolysis efficiency. To better understand the affection of ethanol stress toward the metabolism and growth of *A. oryzae*, this study first investigated the intracellular changes of fatty acid and amino acid compositions in response to ethanol adversity and then uncovered the different expression profiles of desaturase genes involved in fatty acid desaturation. Our study aimed to elucidate the adaptive mechanism of *A. oryzae* toward ethanol stress and to improve the ethanol tolerance of *A. oryzae* strains.

## 2. Materials and Methods

### 2.1. Effects of Ethanol Stress on A. oryzae

The *A. oryzae* 3.042 strain (CICC 40092) exhibits unique characteristics (e.g., amylase and protease production, and stronger potential for environmental stress resistance) [24]. This strain is qualified with the typical characteristics of strains used for rice wine brewing, and was therefore selected for the preliminary exploration of the growth response to ethanol treatment. The basal growth medium selected for the experiments was potato dextrose agar (PDA) medium. To prepare the *A. oryzae* inoculum, some conidia were inoculated on a PDA plate and incubated for 3 days. Conidia were then collected and suspended in sterile water with 0.05% Tween-80, and the concentration of conidia was determined using a hemocytometer. In order to investigate the response to ethanol stress, solid PDA plates with final ethanol concentrations of 2%, 4%, and 6% were employed as the treatment, and meanwhile a medium in absence of ethanol was set as the control. The ethanol tolerance of the *A. oryzae* 3.042 strain was first analyzed by spot dilution assay on a solid PDA medium. Briefly, 2 μL of each 10-fold dilution was spotted onto a solid plate cultivated at 30 °C, and the mycelioid colonies were observed after 48 h of incubation. In addition, a freshly prepared suspension with 1 × 10^7^ conidia was inoculated onto a plate covered with cellophane (Solarbio, Beijing, China), and all cultures were incubated at 30 °C. The fungal mycelia were collected at intervals of 12 h from the first 24 h of incubation to 72 h. After that, the fungal mycelia were dried overnight at 60 °C, and their biomass was weighed. To investigate the effect of ethanol on conidia formation, a parallel experiment was performed to calculate the conidia number at each time point. For each plate, the conidia were suspended in 10 ml of sterile water, and the concentration was determined using a hemocytometer. To determine the composition of intracellular fatty acids, samples were collected in the same manner. For each assay, three replications were performed at each time point.

### 2.2. Transmission Electron Microscopy (TEM) Analysis

To examine affection when exposed to ethanol stress, the morphological changes in organelles were investigated by TEM analysis. After 36 h of incubation on a PDA plate with or without ethanol, some fungal mycelia were sampled and rinsed with 0.1 M of phosphate buffer solution (PBS; pH 7.2). Samples were then fixed in a mixture of 2% glutaraldehyde and 4% paraformaldehyde in 0.1 M of PBS (pH 7.2) overnight at room temperature. After washing three times with PBS (10 min each), samples were postfixed in 1% osmium tetroxide in PBS for 2 h. Then, the samples were briefly washed with the same PBS, dehydrated in an ethanol series (30%–100%), and finally embedded in London Resin (LR) white resin (Taab, Aldermaston, Berks, UK) for polymerization. Ultrathin sections 60–80 nm thick were cut using a diamond knife on a Reichert Ultracut ultramicrotome (Reichert Company, Vienna, Austria). Subsequently, these sections were observed under a Hitachi H-7500 TEM (Hitachi Ltd., Tokyo, Japan).

### 2.3. Determination of Cellular Fatty Acid Content

Total lipids were extracted from whole-cell homogenates and methylated as previously described [25]. In brief, fungal mycelia were collected, washed with distilled water, and freeze-dried. The mycelia were then powdered, weighed, and subjected to lipid extraction. The lipid extract was pretreated by acid-catalyzed esterification (incubating in anhydrous methanol with 2% H_2_SO_4_ at 70 °C for 2 h) to obtain fatty acid methyl esters (FAMEs). The FAME components were separated and analyzed using a coupled QP2010 gas chromatography–mass spectrometry (GC–MS) system (Shimadzu, Kyoto, Japan) equipped with a BPX70 capillary polar column (30 m × 0.22 mm i.d., film thickness of 0.25 μm; SGE, Austin, TX, USA). FAMEs were identified based on comparisons of their mass spectra to a spectrum database. The relative amounts of individual components were measured based on the peak area, and the percentage of UFAs was also calculated.

### 2.4. Effects of Ethanol Stress on Gene Transcription

To investigate the effects of ethanol treatment on gene transcription, qRT-PCR analysis was conducted on genes involved in linoleic acid (Δ^9,12^C18:2) biosynthesis. Total RNA was extracted from the fungal mycelia using a fungal RNA kit (Omega Bio-tek, Norcross, USA). First-strand cDNA was synthesized using a PrimeScript RT reagent kit with gDNA Eraser (Takara, Dalian, China). The qRT-PCR analysis was performed using a CFX96 Real-Time PCR Detection System (Bio-Rad) and a SYBR Green SuperReal PreMix Plus (TianGen, Beijing, China). Two reference genes, *18S rRNA* and *GADPH*, were employed to normalize the target gene expression and to rectify the sample-to-sample discrepancy. The amplification efficiencies of target and reference genes were calculated using a gradient dilution of templates, and the results showed that the amplification efficiencies for target and reference genes were similar and close to 100%. A blank control without template was included in each experiment. Each reaction for one sample was performed in triplicate, and three biological replicates were performed. The comparative 2^−ΔΔCT^ method was employed to calculate the relative expression between samples.

### 2.5. Determination of Intracellular Free Amino Acids

For determination of the intracellular amino acids, *A. oryzae* was cultured in PDA medium containing different concentrations of ethanol. After 36 h of cultivation, the mycelia were harvested and freeze-dried overnight. Subsequently, 0.1 mg of mycelia was treated with liquid nitrogen grinding plus ultrasonic crushing. After that, the cell suspension was collected by centrifugation to remove the cell debris, and the supernatant was transferred for the amino acid assay by high-performance liquid chromatography (HPLC; 1260 series, Agilent technology) after derivatization. The method for online pre-column derivatization was operated according to Agilent guidance, and orthophthalaldehyde (OPA) and 9-formic acid methyl ester of fluorine chlorine (FMOC) were selected as the derivatization agents. The column used for the HPLC analysis was a ZORBAX Eclipse-AAA column at a flow rate of 2 mL/min. The FMOC-derived amino acids were detected at 262 nm, whereas the OPA-derived amino acids were detected at 338 nm.

### 2.6. Data Analysis

All of the data in this study are presented as mean ± SE. Significant differences were determined through a one-tailed Student’s *t*-test or a one-way analysis of variance (ANOVA) followed by Tukey’s honestly significance difference (HSD) test for mean comparison. All statistical analyses were operated with SAS 9.20 software (SAS Institute, Cary, NC, USA).

## 3. Results

### 3.1. Effect of Ethanol Stress on Growth and Development of A. oryzae

To determine the effect of ethanol on the fungal growth of *A. oryzae*, serial dilutions of spores were spotted on solid medium with different concentrations of ethanol applications. After incubation for 48 h, the morphological characteristics of *A. oryzae* mycelioid colonies showed significant differences between the ethanol treatment and the control. Compared to the control, the vegetative mycelia appeared much smaller in 4% ethanol application, and the growth of mycelia was completely inhibited upon 6% ethanol addition (Figure 1). A similar result was obtained in solid medium cultivation, wherein the ethanol treatments severely delayed mycelial development and conidia formation under ethanol stress (Figure 2). The dry biomass of *A. oryzae* mycelia cultivated in solid medium revealed an obvious difference in response to ethanol treatment, particularly between 4% ethanol treatment and the control. More notably, a significant difference was observed during the first 48 h of incubation. Taken together, these results indicated that ethanol treatment significantly suppressed the growth of fungal mycelioid colonies and that the suppressive effect increased with rising concentrations of ethanol.

Because no conidia were produced in liquid culture, a solid medium was employed to test the effect of ethanol treatment on the conidia formation of *A. oryzae*. The result indicated that conidia production was significantly inhibited upon ethanol treatment relative to the control, and the suppressive effect became more obvious with increasing levels of ethanol treatment.

### 3.2. Morphological Changes under Ethanol Treatment

The morphological characteristics after ethanol treatment were determined by transmission electron microscopy (TEM) analysis (Figure 3). In the absence of ethanol, typical morphologies were observed for mitochondria, vacuoles, and nuclei in *A. oryzae* hyphae. In non-ethanol conditions, mitochondria were distributed evenly in the cytoplasm, and vacuoles were intact and relatively small and dispersed. When *A. oryzae* hyphae were exposed to 4% ethanol, the scattered vacuoles became increasingly aggregated, and even worse, they formed a large and single vacuole. Meanwhile, the nuclei appeared highly swollen and lengthened in the hyphae under ethanol treatment. Mitochondrial structures appeared swollen with less structured cristae. Taken together, these results suggested that ethanol uptake resulted in membrane damage and further impaired the structure and function of organelles.

### 3.3. Determination of Intracellular Fatty Acid Compositions

The alterations in membrane compositions play a crucial role in the stress response system, which enables the strains to thrive against ethanol pressure [7]. The relationship between ethanol tolerance and an increased degree of fatty acid unsaturation of membrane lipids has been well documented in *S. cerevisiae* [9]. In the present study, the fatty acid content profiles of *A. oryzae* grown at different levels of ethanol were quantified. Our results revealed that the contents of 18- and 16-carbon chain length fatty acids comprised the predominant fatty acids, accounting for approximately 76.7% and 18.7% of total fatty acids, respectively (Figure 4). The major fatty acids in *A. oryzae* were oleic acid (Δ^9^C18:1), linoleic acid (Δ^9,12^C18:2), stearic acid (C18:0), and palmitic acid (C16:0). When subjected to 4% ethanol, a major change was observed in the degree of fatty acid unsaturation 36 h post-treatment, and the intracellular content of unsaturated fatty acids increased by 12.8% in adaption to ethanol conditions (Figure 4). The results also showed that the percentage of unsaturated fatty acids increased with rising concentrations of ethanol. However, the discrepancy in fatty acid unsaturation between different treatment groups became unobtrusive 84 h post-treatment.

In *A. oryzae* cultures subjected to 4% ethanol, a remarkable change was observed in the major fatty acid components of C18, i.e., a decrease of 12.32% in stearic acid (C18:0) content, and accompanied by an obvious 12.62% increase in linoleic acid (Δ^9,12^C18:2) was observed, compared to the control grown without ethanol (Appendix A). Apart from this, the content of oleic acid (Δ^9^C18:1) remained steady. In addition, the content of linoleic acid exhibited a positive correlation with the rising concentrations of ethanol treatment 36 h post-treatment (Figure 5A). However, no significant difference was detected in linoleic acid content at 84 h in the different treatments. Combined with the Kyoto Encyclopedia of Genes and Genomes (KEGG) pathway analysis, it was assumed that stearic acid was converted into linoleic acid through an intermediate (oleic acid) in *A. oryzae* cells responding to ethanol treatment (Figure 5B).

### 3.4. Differential Expression of Desaturases in Linoleic Acid Biosynthesis

To further gain insight into the roles of linoleic acid in ethanol tolerance, the transcriptional levels of Δ^9^-desaturase (D9d) and Δ^12^-desaturase (D12d) involved in linoleic acid biosynthesis were examined in response to ethanol treatment. In the biosynthesis of linoleic acid, D9d acted as a rate-limiting enzyme, converting stearic acid into oleic acid. The expression of D9d1 was upregulated by 2.6-fold 36 h post-4% ethanol treatment (Figure 6). However, no significant difference was detected in D9d1 expression between the treated and control groups at 84 h. Meanwhile, ethanol treatment resulted in a decrease in D9d2 expression at 36 h, and conversely, no significant difference was observed in D9d2 expression at 84 h post-treatment. D12d, which is responsible for the conversion of oleic acid into linoleic acid, showed significantly increased expression when exposed to 4% ethanol treatment (*p* < 0.01), whereas this tendency became indistinct 84 h post-treatment.

### 3.5. Determination of Intracellular Free Amino Acids

To elucidate the protective mechanism of amino acids in *A. oryzae* cells under ethanol stress, the intracellular free amino acids were quantified. After 36 h cultivation in a 4% ethanol condition, a significant increase was observed in the intracellular accumulation of arginine (Arg) and asparagine (Asn) (Figure 7). Compared to the control group, the free arginine composition increased by 50.8% under 4% ethanol treatment. Meanwhile, no apparent changes were observed in other types of amino acids.

## 4. Discussion

During rice wine brewing, the accumulating ethanol produced by sake yeast undoubtedly becomes an inhibitor of the growth and fermentation performance of *A. oryzae* cells. In our preliminary experiment, ethanol treatment exhibited a significant inhibition of mycelium growth and conidia formation in *A. oryzae*, and this suppressive effect showed a positive correlation with ethanol concentrations. Similarly, in budding yeast, it has been well documented that high levels of ethanol impair cell growth and viability, affecting their fermentation capability in bioethanol production [9,26]. Our results revealed that *A. oryzae* cells reached a maximum tolerance to ethanol at 6% (v/v), whereas sake yeast can produce ethanol until the ethanol concentration reaches approximately 20% (v/v) [3]. The discrepancy in ethanol tolerance between yeast and koji mold results in the advanced autolysis of *A. oryzae* mycelia and a decline in the hydrolysis of starch during rice wine brewing. Obviously, an enhancement of the ethanol tolerance of *A. oryzae* would undoubtedly contribute to long-term fermentation. In addition to defects in cell growth and conidia formation, ethanol also triggered internal cellular perturbations, such as irregular nuclei and the aggregation of scattered vacuoles in *A. oryzae* cells. Similar morphologic characteristics have also been observed in yeast when exposed to ethanol, indicating the subhealth status of organelles [27]. Although the physiological function of vacuolar morphological changes remains unclear, the aggregation of vacuoles indicates an inhibition of cell division and a decline in cell viability. 

The cellular membrane is the primary target that suffers from the injurious effects of ethanol, and it has been reported that the membrane and cell wall structures undergo significant remodeling processes in response to ethanol stress [7,9,27,28,29]. So far, a detailed study of ethanol tolerance mechanisms has been well documented in ethanologenic yeast. It was reported that the ethanol tolerance in yeast was largely attributed to the incorporation of oleic acid into lipid membranes, counteracting the fluidizing effects of ethanol toward the membrane [9]. Similarly, global metabolite profiling in ethanol-tolerant yeast strains has revealed compositional changes in cell membranes, with an increased ratio of monounsaturated fatty acids [5]. The predominant fatty acids in *A. oryzae* are palmitic acid (C16:0), stearic acid (C18:0), oleic acid (Δ^9^C18:1), and linoleic acid (Δ^9,12^C18:2), accounting for 93.4% of the total. When exposed to ethanol perturbation, a remarkable increase was detected in fatty acid unsaturation, and a large proportion (approximately 12%) of stearic acid was converted into linoleic acid in *A. oryzae* cells. In agreement with this, an increase in the saturation degree of fatty acids counteracted the elevated membrane fluidity caused by ethanol and maintained membrane stability in *Arthrobacter simplex* [30]. In fact, the degree of fatty acid unsaturation is tightly correlated with membrane fluidity, and the upregulation of unsaturation contributes to maintaining the membranes in an appropriate fluid state [28]. To date, lipid modification through activation of fatty acid desaturation in response to abiotic stress has been widely documented, ranging from bacteria and fungi to plants [9,29,31,32,33]. A high proportion of cell membrane lipid moieties are comprised of unsaturated fatty acids that are synthesized from saturated fatty acids through specific enzymes [34,35]. When subjected to ethanol treatment, our results also revealed the activation of desaturases involved in linoleic acid synthesis, of which d9d1 and d12d encountered a significant and transient upregulation in transcription levels. This suggests that ethanol adversity resulted in a transient induction of stress protective gene expression, and the metabolism of unsaturated fatty acids was more active in order to maintain membrane integrity [36]. The induced expression of fatty acid desaturases (FADs) for fatty acid desaturation represented a tight correlation between transcriptional regulation and metabolite changes in lipid metabolism responding to ethanol stress. Similarly, the exposure of yeast or *A. oryzae* cells to saline stress results in a transient induction of the expression of stress protective genes, in which the pathway of fatty acid biosynthesis becomes active [29,37]. However, the physical phenomena behind this correlation are likely to be complex. It has also been reported that the expressions of genes associated with the stabilization or refolding of denatured proteins, such as those encoding heat shock proteins (HSPs) and trehalose metabolic enzymes, were rapidly upregulated upon ethanol exposure in yeast [38].

Our results also revealed an increase in intracellular content of arginine after ethanol treatment. Similarly, in yeast, either extracellular arginine addition or the overexpression of arginine biosynthesis genes enhances yeast tolerance to ethanol stress, and a further study revealed that arginine contributes to the integrity of the cell wall and cytoplasma membrane [27,39]. A similar result in *Escherichia coli* reported that supplementary arginine in culture protected cells against hydrogen peroxide-induced oxidative perturbation [40]. In *Candida glabrata* under hyperosmotic conditions, the transcription of genes encoding enzymes for arginine biosynthesis were increased, whereas that of genes encoding enzymes for arginine degradation were decreased, and the overexpression of two key enzymes in the arginine synthesis pathway promoted cell growth [41]. These studies strongly support the assumption that arginine can counteract the cellular damage caused by ethanol treatment by maintaining the intracellular physiological environment as relatively stable.

In conclusion, our study uncovered new aspects of the dynamic changes in *A. oryzae* mycelium growth and conidia formation over time under ethanol stress, characterized the upregulation of linoleic acid synthesis, and improved the understanding of mechanisms involved in the ethanol tolerance of *A. oryzae*. A clear insight into fatty acid metabolism will contribute to a better understanding of ethanol tolerance mechanisms and facilitate the construction of industrial strains with a higher ethanol tolerance. Recent advances in the genetic manipulation of *A. oryzae* using *Agrobacterium tumefaciens*-mediated transformation (ATMT) could facilitate this goal [42,43].

## Figures and Tables

**Figure 1 microorganisms-07-00158-f001:**
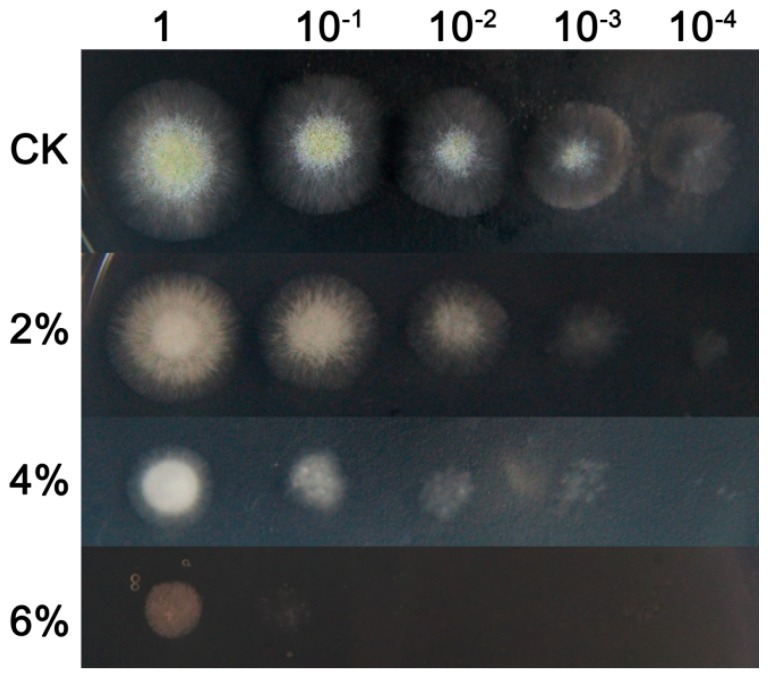
The inhibitory effect of ethanol variations at different concentrations (from 0% to 6%) on *Aspergillus oryzae* colonies at 48 h post-treatment. The suppression effect was determined in potato dextrose agar (PDA) solid medium by inoculating the serial dilutions of *A. oryzae* spores (from 1 to 10^−4^).

**Figure 2 microorganisms-07-00158-f002:**
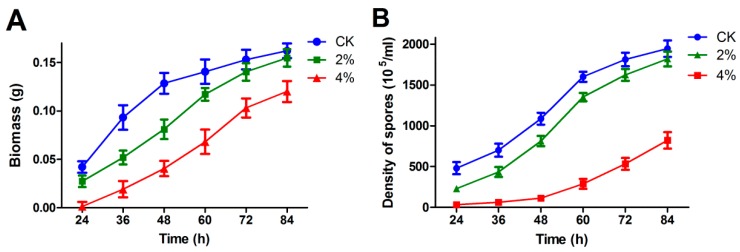
The mycelia biomass and conidia number of *Aspergillus oryzae* in response to ethanol stress. (**A**) The dried biomass of *A. oryzae* mycelia under ethanol conditions from 24 to 84 h post-treatment. (**B**) The number of *A. oryzae* conidia under different concentrations of ethanol treatment.

**Figure 3 microorganisms-07-00158-f003:**
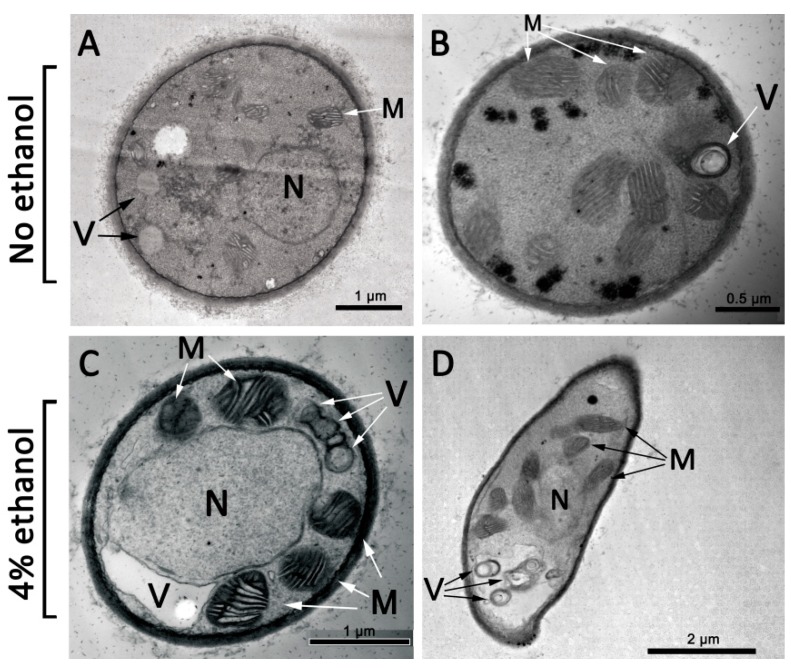
Transmission electron microscopy analysis of the adaptive response of *A. oryzae* to ethanol stress. Under conditions of 4% ethanol treatment, the segregated vacuoles became increasingly aggregated, and the cellular nuclei appeared swollen and lengthened. (**A,B**) *A. oryzae* cells without ethanol. (**C,D**) *A. oryzae* cells under 4% ethanol. M: mitochondria; N: nucleus; V: vacuole.

**Figure 4 microorganisms-07-00158-f004:**
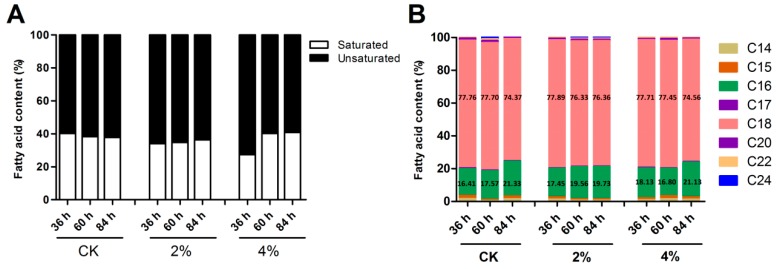
The degree of fatty acid unsaturation in different concentrations of ethanol treatments. (**A**) The changes in the content of unsaturated fatty acids in response to ethanol treatment. (**B**) The intracellular content of different types of fatty acids under varying concentrations of ethanol treatments.

**Figure 5 microorganisms-07-00158-f005:**
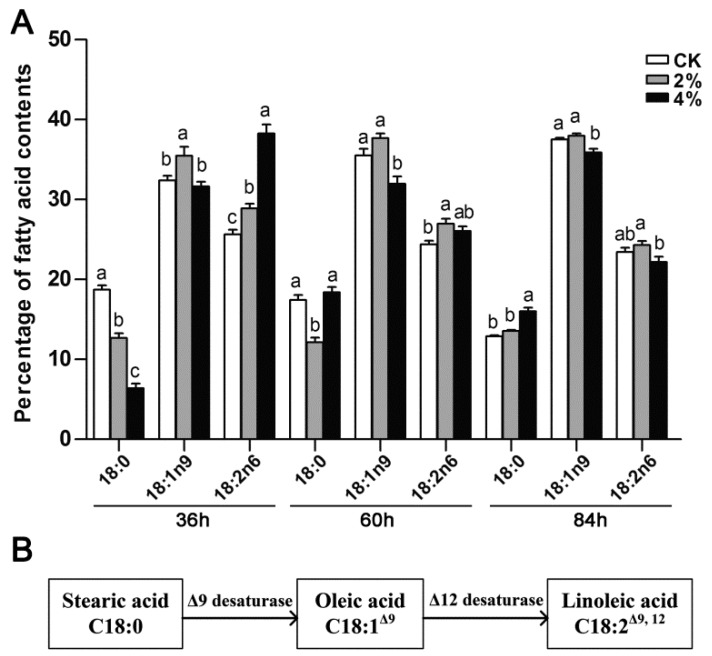
The changes of C18 fatty acid compositions in response to ethanol stress at 36, 60, and 84 h post-treatment (**A**). The elucidation of the linoleic acid biosynthesis pathway in *Aspergillus oryzae* according to KEGG pathway analysis (**B**). The bars represent the average (±SE) of biological repeats. Different letters above the bars indicate significant differences (*p* < 0.05).

**Figure 6 microorganisms-07-00158-f006:**
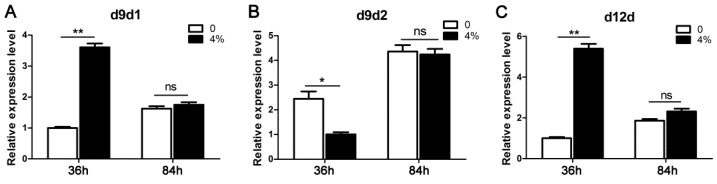
The effects of ethanol treatment on the expressions of Δ^9^-desaturase (D9d) and Δ^12^-desaturase (D12d) in linoleic acid biosynthesis 36 and 84 h post-treatment. Asterisks indicate statistically significant differences between groups (Student’s *t*-test): * denotes *p* < 0.05; ** denotes *p* < 0.01; ns: no significant difference.

**Figure 7 microorganisms-07-00158-f007:**
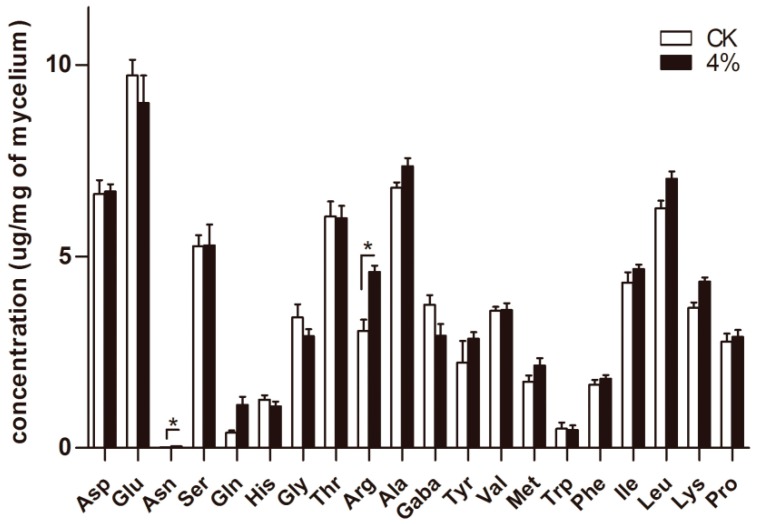
Alterations in the intracellular amino acid content under the condition of 4% ethanol at 36 h post-treatment. The bars represent the average (±SE) of biological repeats. Asterisks indicate statistically significant differences between groups (*t*-test): * denotes *p* < 0.05.

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
