# Peer review of "Modulation of Fatty Acid Composition of Aspergillus oryzae in Response to Ethanol Stress"

_microorganisms, 2019, doi:10.3390/microorganisms7060158_

Round 1

Reviewer 1 Report

Ma et al. report the phenotypes of Aspergillus oryzae under ethanol stress and attempted to study the mechanisms responding to ethanol stress. However, it is not clear what the conclusion/information that the authors want to convey. The paper needs to be reorganized.

Additional questions/comments:

In the introduction section

Introduction of the molecular mechanisms of ethanol tolerance are not enough. The authors need to provide more information/background on it. Most of the work that the authors cited are out of date, newer studies need to be present. For example, very recent study (Morard et al, Front Genet. 2019 ) showed that aneuploidy could also be a mechanism to adapt ethanol stress. There is a much recent review about ethanol tolerance in yeast (Snoek, T., Verstrepen, K. J., and Voordeckers, K. (2016). How do yeast cells become tolerant to high ethanol concentrations? Curr. Genet. 62, 475–480.)

L31 produced

L49 it was reported

L51 proven

L54 pro1 should be in lowercase as it is a mutation

L59 advantage

L61 towards

In the results section,

What’s the rationale of testing ethanol concentration from 2% - 6%? Are those concentrations common during fermentation? Why throughout of the study, the authors keep using 4% ethanol?

Why 3.6 data analysis was put here? Shouldn’t it put in the methods section?

Except unit 3.1, there are no conclusions was made for other units. I don’t know what the information that the authors want to convey.

Fig.3 Scale bar of TEM should be labeled. No description of panel A-D. Please revisit.

Author Response

Dear editor,

On behalf of my co-authors, we thank you very much for giving us an opportunity to revise our manuscript. We appreciate the reviewers very much for their positive and constructive comments on our manuscript. A professional English editing service offered by native speaker was employed to polish the language. Based on these suggestions, we tried our best to edit this article. The manuscript was modified systemically using the track change mode. The corrections are listed below point by point.

1. Introduction of the molecular mechanisms of ethanol tolerance are not enough. The authors need to provide more information/background on it. Most of the work that the authors cited are out of date, newer studies need to be present. For example, very recent study (Morard et al, Front Genet. 2019 ) showed that aneuploidy could also be a mechanism to adapt ethanol stress. There is a much recent review about ethanol tolerance in yeast (Snoek, T., Verstrepen, K. J., and Voordeckers, K. (2016). How do yeast cells become tolerant to high ethanol concentrations? Curr. Genet. 62, 475–480.)

Respond: Thanks for your advise. In the present version, the latest studies regarding the mechanisms of ethanol tolerance have been supplemented in the section of introduction (including the effect of membrane ergosterol contents, heat shock proteins (HSPs), trehalose, and aneuploidy in chromosome III) as follows, “Similarly, Bacillus subtilis and Clostridium acetobutylicum elevated the content of saturated fatty acids under ethanol condition [Weber and de Bont 1996]. In addition to unsaturated fatty acids (UFAs), the membrane ergosterol contents also contribute to ethanol tolerance, possibly due to its effect on enhancing membrane rigidity [Abe 2009]. Consistent with this assumption, the mutants defective in ergosterol biosynthesis were extremely susceptible to ethanol adversity [Auesukaree et al.,  2009; Teixeira et al., 2009]. To cope with the effect of ethanol, heat shock proteins (HSPs) were required to assist the folding and maintenance of newly assembled proteins, and the suppression of protein aggregates [Verghese et al., 2012]. In agreement, the yeast mutants of HSP12, HSP30, or HSP104 genes became hypersensitive to ethanol [Teixeira et al., 2009; Yoshikawa et al., 2009]. To protect yeast cells from protein denaturation caused by ethanol, trehalose cooperated with HSPs to stabilize protein structure through its function in protein binding and reduction of water activity [Jain et al., 2009]. Higher intracellular trehalose accumulation by overexpression of trehalose-6-phosphate synthase enhanced the ethanol tolerance in the engineered yeast strain [Divate et al., 2016].”

“The comparative genomics analysis of five yeast strains with different ethanol tolerances revealed that chromosome III was important for the ethanol stress response, and this aneuploidy contributed to the rapid response to ethanol stress; and conversely, the removal of extra chromosome III copy in an ethanol-tolerant strain strongly weakened its tolerance [Morard et al.,  2019].”

2.

L31 produced

L49 it was reported

L51 proven

L54 pro1 should be in lowercase as it is a mutation

L59 advantage

L61 towards

Respond: Thanks for your valuable advise. These mistakes have been corrected.

In the results section,

3. What’s the rationale of testing ethanol concentration from 2% - 6%? Are those concentrations common during fermentation? Why throughout of the study, the authors keep using 4% ethanol?

Respond: Thanks for your suggestion. In our preliminary experiment, we investigated the ethanol tolerance of A. oryzae by spot dilution assay on solid PDA medium. Briefly, the solid PDA plates with final ethanol concentrations ranging from 2% to 10% were prepared (described in the section of method), and serial dilutions of A. oryzae spores were inoculated on PDA plate and incubated for 3 days. The resulted revealed that the growth of mycelia was completely inhibited upon 6% ethanol addition (Figure 1). Many studies have documened the method of spot dilution assay to test the effect of exogenous additions on the viability and growth of cells, such as the effect of exogenous ergosterol on the viability of D-limonene treated yeast cells (Liu et al., 2013).

As we described in the section of introduction and discussion, ‘The traditional alcoholic beverage of rice wine is produced through parallel fermentation, with a simultaneous saccharification of rice by the koji mould A. oryzae and ethanol fermentation by the sake yeast Saccharomyces cerevisiae’ (Kotaka et al., 2008). In general, sake yeast can produce ethanol until the ethanol concentration reaches approximately 20% (v/v) (Kotaka et al., 2008). Due to the maximum ethanol tolerance of 6%, the A. oryzae mycelium begins to autolyze in advance during fermentation, which enhances the volatile flavor of fermented productions (Xu et al., 2016).

Our results revealed that 4% ethanol exhibited the obvious inhibitive effect towards the mycelial growth while the suppressive effect becomes unclear in 2% ethanol. Meanwhile, A. oryzae cells reached the maximum tolerance of ethanol at 6%. Considering this issue, the 4% ethanol was finally determined in the study.

Kotaka A, Bando H, et al. Direct Ethanol Production from Barley β-Glucan by Sake Yeast Displaying Aspergillus oryzae β-Glucosidase and Endoglucanase. J Biosci Bioeng. 2008, 105(6): 622-627. 

Liu j, Zhu Y, et al. Exogenous ergosterol protects Saccharomyces cerevisiae from D-limonene stress. J Appl Microbiol. 2013, 114(2): 482-491.

Xu N, Liu Y, et al. Autolysis of Aspergillus oryzae mycelium and effect on volatile flavor compounds of soy sauce. J Food Sci. 2016, 81(8):1883-1890.

4. Why 3.6 data analysis was put here? Shouldn’t it put in the methods section?

Respond: Thanks for your advise. This section has been adjusted to the section of methods.

5. Except unit 3.1, there are no conclusions was made for other units. I don’t know what the information that the authors want to convey.

Respond: Thanks for your advise. The conclusion for 3.2 was depicted as follows, ‘Taken together, these results suggested that ethanol uptake resulted in membrane damage and further impaired the structure and function of organelles. (for 3.2)’.

For part of 3.3, the conclusion is “When subjected to 4% ethanol, a major change was observed in the degree of fatty acid unsaturation at 36 h post-treatment, and the intracellular content of unsaturated fatty acids was increased by 12.8% in adaption to ethanol conditions”.

For part of 3.4, the conclusion is “a large proportion of stearic acid was converted to linoleic acid, and the expressions of related fatty acides desaturases were activated.”.

For part of 3.5, the conclusion is “In 4% ethanol condition, a significant increase was observed in the intracellular accumulation of arginine (Arg) and asparagine (Asn)”.

In conclusion, our study uncovered new aspects of the dynamic changes in A. oryzae mycelium growth and conidia formation over time under ethanol stress, characterized the up-regulation of linoleic acid synthesis, and improved the understanding of mechanisms involved in ethanol tolerance of A. oryzae. A clear insight into fatty acid metabolism will contribute to a better understanding of ethanol-tolerance mechanisms and facilitate the construction of industrial strains with higher ethanol tolerance.

6. Fig.3 Scale bar of TEM should be labeled. No description of panel A-D. Please revisit.

Respond: Thanks for your advise. The plotting scale would be enlarged and become clear in the current version. The elucidation for figure A, B, C and D has been supplemented in the figure legend in the current version.  

Reviewer 2 Report

I encourage the authors to make some minor revisions to improve the quality of the manuscript.

Line 23. I suggest to change the sentence by including the word “understanding” as follows: Our results therefore improve the understanding of the ethanol adaptation mechanisms in A. oryzae. See if this is the idea you want to express.

Line 70. Change to: characteristics

Line 90. Affectation instead of affection or, modify the beginning of the phrase.

Line 92. I would suggest to include the meaning of the abbreviation PBS for the first time (Phosphate buffer saline); it would be clarifying for those not knowing the meaning.

Line 93. Should it be 0.1M PBS?

Line 143. Change to: characteristics

Line 157. Change to: By inoculating the serial dilution

Line 167. Change to: characteristics

Figure 3. The scales in the images are hard to read, do they have the same scale? And furthermore, the letters, A, B, C and D refer to any particular case we should know about?

Line 181. Erase second end point.

Line 264. Change to: morphological characteristics were also observed

Line 283-284. Rewrite the sentence to clarify the idea. You meant “a high proportion of cell membrane unsaturated fatty acids are synthesized from saturated fatty acids through specific enzymes”?

Line 304. Change to: relatively

Author Response

On behalf of my co-authors, we thank you very much for giving us an opportunity to revise our manuscript, we appreciate the editor very much for their positive and constructive comments and suggestions on our manuscript. What’s more, thanks for the professional suggestion from the reviewers, the reviewers showed his or her responsibility, professional knowledge and patience to this manuscript. Based on these suggestions, we tried our best to edit this article. A professional English editing service offered by native speaker was employed to polish the language. The manuscript was modified systemically using the track change mode. The corrections are listed below point by point. 

Line 23. I suggest to change the sentence by including the word “understanding” as follows: Our results therefore improve the understanding of the ethanol adaptation mechanisms in A. oryzae. See if this is the idea you want to express.

Respond: Thanks for your suggestion. The sentence here has been modified as advised.

Line 70. Change to: characteristics

Respond: Thanks for your suggestion. This has been modified in the current version.

Line 90. Affectation instead of affection or, modify the beginning of the phrase.

Respond: Thanks for your suggestion. This has been modified.

Line 92. I would suggest to include the meaning of the abbreviation PBS for the first time (Phosphate buffer saline); it would be clarifying for those not knowing the meaning.

Respond: Thanks for your suggestion. This has been modified.

Line 93. Should it be 0.1M PBS?

Respond: Thanks for your correction, and this has been corrected.

Line 143. Change to: characteristics

Line 157. Change to: By inoculating the serial dilution

Respond: Thanks for your suggestion. This has been modified as advised.

Line 167. Change to: characteristics

Respond: Thanks for your suggestion. This has been modified.

Figure 3. The scales in the images are hard to read, do they have the same scale? And furthermore, the letters, A, B, C and D refer to any particular case we should know about?Respond: Thanks for your advise. The plotting scale would be enlarged and become clear in the current version. The elucidation for figure A, B, C and D has been supplemented in current version.  

Line 181. Erase second end point.

Line 264. Change to: morphological characteristics were also observed

Respond: Thanks for your suggestion. This has been modified.

Line 283-284. Rewrite the sentence to clarify the idea. You meant “a high proportion of cell membrane unsaturated fatty acids are synthesized from saturated fatty acids through specific enzymes”?

Respond: This sentecne has been modified as follows, ‘A high proportion of cell membrane lipid moieties are comprised of unsaturated fatty acids that are synthesized from the saturated fatty acids through specific enzymes’.

 Line 304. Change to: relatively

Respond: Thanks for your suggestion. This has been modified.

Round 2

Reviewer 1 Report

The authors generally answer all of my questions. I don't have additional comments, except for some minor issues described below:

L12, L13, mould is a British spelling, whereas mold is an American. You should keep one type rather than messing them up throughout the manuscript.

L19, the plural form of nucleus is nuclei.